# Beef from Calves Finished with a Diet Based on Concentrate Rich in Agro-Industrial By-Products: Acceptability and Quality Label Preferences in Spanish Meat Consumers

**DOI:** 10.3390/ani12010006

**Published:** 2021-12-21

**Authors:** Elena Angón, Francisco Requena, Javier Caballero-Villalobos, Miguel Cantarero-Aparicio, Andrés Luís Martínez-Marín, José Manuel Perea

**Affiliations:** 1Departamento de Producción Animal, Universidad de Córdoba, 14071 Cordoba, Spain; eangon@uco.es (E.A.); javier.caballero@uco.es (J.C.-V.); t42caapm@uco.es (M.C.-A.); pa1martm@uco.es (A.L.M.-M.); jmperea@uco.es (J.M.P.); 2Departamento de Biología Celular, Fisiología e Inmunología, Universidad de Córdoba, 14071 Cordoba, Spain

**Keywords:** aging, animal welfare, beef, by-products, conjoint, consumer, finishing heifers, meat, organic farming, quality label

## Abstract

**Simple Summary:**

The replacement of cereals with human-inedible biomass is a strategic method to reduce food–feed competition, mitigate the environmental impact of livestock, and reduce production costs. This study proves that the fattening of calves with a diet rich in human-inedible fibrous local agro-industrial by-products from southern Spain improves the color, flavor, and tenderness of meat, increasing its acceptance by regular meat consumers. This study also analyzed the importance of quality labels in the formation of preferences of Spanish consumers, finding that origin, price, and animal welfare certification are the most important attributes. Spanish consumers seem to prefer meat with the lowest possible price, of national origin, and with the highest possible number of quality labels.

**Abstract:**

Conjoint analysis was used to estimate the relative importance of some of the main extrinsic attributes and quality labels of beef in three Spanish cities (Córdoba, Marbella, and Santa Pola) in a study performed with 300 individuals. Consumers were segmented according to their frequency of consumption. Willingness to pay for different meats was also calculated from the conjoint analysis results. Consumer liking of beef that had been finished with an alternative concentrate rich in agro-industrial by-products and aged for three different durations as compared to conventionally finished beef was also evaluated using the same consumers. The most important attribute for Spanish consumers was the price (28%), followed by origin (25%), animal welfare certification (19%), protected geographical indication (14%), and organic agriculture certification (14%). Most consumers preferred beef from Spain at the lowest possible price and with the highest number of quality labels. Consumers were willing to pay a premium of 1.49, 3.61, and 5.53 EUR over 14 EUR/kg for organic certification, protected geographical indication, and animal welfare certification, respectively. Sensory analysis revealed that, for regular consumers, beef finished with an alternative concentrate rich in agro-industrial by-products offered several hedonic advantages (color, flavor, and tenderness) when compared to beef finished using a conventional diet, while occasional consumers did not find any difference between the two kinds of meat.

## 1. Introduction

The EU Commission recently established a priority target for 2030 to reduce per capita food waste by 50%, following the “Farm to Fork” strategy (F2F) established within the frame of the European Green Deal [1]. For this reason, crop diversification represents a key pillar in the agroecological transition due to its positive effects on productivity, soil quality and fertility, resistance to plagues and plant diseases, reduced use of fertilizers, and decreased environmental stress [2]. Sustainability, based on the availability of resources, functionality, and the morality of use, aims to guarantee a supply of safe and quality foods through the development of competitive and eco-efficient distribution chains [3,4]. The use of by-products and agro-industrial residues for animal feed is considered within this context [5,6]. Consumers are increasingly aware of the benefits that these practices have for the ecosystem, in addition to reduced exposure to polluting substances that may pose a risk to health [7,8]. Although there is a wide variety of by-products and agro-industrial residues that can be used for feeding livestock, there is scarce information on how they can impact food quality and safety [9]. Some studies have evidenced risks for human health related to the contamination and transfer of biological and chemical agents to animal feed and their subsequent entry into the food chain through food from animal sources [10,11]. These risks need to be carefully assessed before authorizing the use of any by-product or agri-food residue in animal feed in order to ensure that food safety and public health are not compromised.

The scarce availability of pastures and forages in Mediterranean areas (arid or semiarid), which is the case for southern Spain, makes the search for local crops and agro-industrial by-products a priority to identify an alternative to the traditional diet for calf finishing, which is based on cereals and imported raw materials [9,12,13]. Diets rich in agro-industrial by-products can affect the growth, carcass, and characteristics of the meat. Some authors have described the effects of these diets on these characteristics [14,15,16], but very few studies have investigated changes in sensory characteristics, such as color or tenderness, which are deemed important to consumers. In this regard, Moreno-Diaz et al. [17] did not observe any negative effects on the productivity or technological features of meat when using a diet with 73.5% agro-industrial by-products from southern Spain when compared to a conventional diet. However, the way that this alternative diet affects sensory acceptance by consumers needs to be studied in depth.

Overall, beef palatability can be attributed to three primary traits: tenderness, juiciness, and flavor. A study [18] reported that these traits accounted for 43.4%, 49.4%, and 7.4%, respectively, of overall palatability. Although its importance depends on various factors, tenderness has been identified as the most important factor of palatability in several studies [19,20,21], and recent investigations have most commonly shown flavor to be the largest factor impacting overall beef eating satisfaction [22,23,24]. Both traits are influenced by intrinsic and extrinsic factors, among which diet [25,26,27,28] and aging [29,30,31,32,33] are notable. The animal feed affects the hygienic, sanitary, nutritional, and sensory characteristics of meat. While the postmortem aging of meat represents a considerable expense to industry, this process improves most attributes of eating quality [33], although the impacts of time to maximize aging on meat quality attributes have not been fully established.

Furthermore, health and environmental concerns are leading Western consumers to reduce their consumption of beef [34,35,36]. For this reason, the meat sector has developed some strategies, among which product differentiation is notable. One of the main ways to differentiate beef is by implementing labels that declare factors that are not directly observable, such as its geographical origin, ethical aspects related to the production process, or parameters related to quality (hygienic, sanitary, nutritional, etc.) [37,38,39]. According to Eurobarometer [40], 8 out of 10 European consumers consider it important for food to include an EU quality guarantee label. Two EU quality marks predominate for beef: protected geographical indication (PGI) and organic agriculture (OA).

Consumers’ inference about meat quality at the point of purchase is based on available intrinsic and extrinsic signs that consumers believe to reflect different product quality attributes [41,42]. Among the meat quality attributes, credibility dimensions such as origin, breed, or production system are noteworthy, and they are expected to gain relative importance in the formation of preferences of meat consumers in the future [43]. Unlike search and experience attributes, credibility attributes cannot be evaluated by typical consumers, not even after consumption. Thus, consumers need information from others to verify the credibility attributes of meat [42,43,44]. If consumers place sufficient confidence in quality brands, these could become the most relevant extrinsic indicators at the point of purchase to verify the credibility attributes of meat quality [43,45]. Consequently, the meat industry has an opportunity in quality brands to better align its products with consumer preferences.

The PGI quality label certifies the geographic origin of meat, emphasizing its reputation, quality and specific characteristics [46]. Spain hosts 11 PGIs for beef, which guarantee the purchase of unique and inimitable meat by consumers. However, little is known about their effectiveness in the preference formation of consumers, as opposed to other meats such as lamb, where the positive impact of the PGI label has been widely proven [47,48].

The OA label applies to meats obtained through processes and farming practices considered more natural, environmentally friendly, and respectful of animal welfare [49]. Although these attributes are well aligned with consumer preferences, organic beef in Spain has not reached a market share comparable to that of the rest of Europe. This is especially evident in southwestern Spain [50], where seasonality and shortage of pastures force the use of organic cereals and more intensive farming practices [13], leading to a high price differential and confusion with traditional systems [51].

Furthermore, several quality labels based on certification schemes have been established in recent years in response to the growing social concern for animal welfare [52,53]. In Spain, these animal welfare (AW) labels are relatively new, but they could acquire great relevance in the market, as they offer the possibility of choosing products obtained through more ethical farming practices but at a more affordable price than that of organic meat [54,55]. However, there is scarce knowledge of the behavior of Spanish consumers towards quality labels, their willingness to pay for a premium and the attributes (both direct and indirect) that determine their purchase decisions [56].

In light of the above, the aims of this study were to determine the relative importance and the willingness to pay for three certified quality labels linked to production systems—PGI, OA and AW—on purchasing decisions of Spanish beef consumers, and to evaluate the sensory acceptability of beef from animals finished with a concentrate rich in fibrous agro-industrial by-products and aged for three time durations.

## 2. Materials and Methods

### 2.1. Material

The experiment was conducted using the longissimus lumborum muscle from 24 crossbred Limousine × Retinta heifers, selected from a batch of 100 cattle according to their diets, and raised and fed as reported by Moreno-Diaz et al. [17]. Briefly, cattle over three months old were randomly assigned to one of two finishing diets: control (CO; cereal-based concentrate plus cereal straw) or alternative (CA; 26% cereals and up to 73.5% agro-industrial by-products, e.g., soybean hulls, hominy feed, corn dried distiller grains, wheat bran, corn gluten feed dehydrated barley sprouts, NaOH-treated wheat straw, camelina meal and camelina husks, plus cereal straw. Once they reached an average of 470 kg live weight (15 months old), heifers were transported to a commercial abattoir for slaughter. At 24 h postmortem, longissimus lumborum muscles were processed into 2.5 cm steaks. Individual steaks were vacuum packed, sent to the laboratory in a refrigerated vehicle, and randomly assigned to one of three postmortem aging periods (in the dark at 2–4 °C for 7, 21 or 28 days), after which they were frozen at −20 °C until being thawed for consumer sensory testing (<2 months of storage).

### 2.2. Subjects

The study was conducted at cooking schools in three cities from the east and south of Spain: two of them were located on the coast (Santa Pola and Marbella), and one was located inland (Córdoba). The facilities were adequately adapted to carry out the sensory tests.

A total of 300 participants (100 per city) were recruited from local communities, including staff and students from the cooking schools who voluntarily agreed to take the test. Recruitment was carried out with the aim of replicating the distribution of the Spanish national population [57]. The demographic characteristics of the sample were similar to those of the population, except for consumers with vocational degrees, who might be slightly overrepresented (Table 1). Participants were assigned into two groups following Żakowska-Biemans et al. [58]: regular consumers are defined as those who consume beef at least once a week (75.4%), and occasional consumers are those who consume beef less regularly (24.6%).

Each participant was asked to complete two tasks: firstly, a sensory analysis and, secondly, a conjoint analysis. For the sensory analysis, to reduce any potential bias, no information on meat type, aging time, or diet was provided. For the conjoint analysis, detailed information was provided on the different attributes under analysis. Sessions were approximately 45 min long, and members of the research team were present to assist participants and answer any queries.

### 2.3. Sensory Analysis

The sensory evaluation was carried out according to previous studies [59,60,61] in a large banquet room, separated from the food preparation area, under fluorescent lighting. Five sensory analysis sessions were conducted in each city, including 20 participants per session. In the blind condition, participants evaluated the acceptability (color, juiciness, tenderness, odor, taste, and overall assessment) of six different samples of beef (two diets: CA and CO; three aging periods: 7, 21 or 28 days). Participants were seated at individual tables that were separated from adjacent ones. They received verbal instructions about how to conduct the test and were requested to score each individual sample on an 8-point category scale (1 = liked very much to 9 = disliked very much). The intermediate level was not included, in accordance with [60]. Samples were presented on white plates, and to avoid order and carry-over effects, they were served following a randomized design [62]. Unsalted crackers and double-distilled deionized water were available to all participants to cleanse their palates between samples [61].

Steak samples were thawed overnight prior to the test at 2–4 °C and then taken out and placed in a room until they reached a temperature of 17–19 °C. After being removed from their packaging, all steaks were cooked in a preheated Gastro M6 oven (IberGastro, Lucena, Spain) at 190 °C until the temperature reached 71 °C at the geometric center. This was monitored by a type-K thermocouple thermometer (HH374 Omega, Omega Engineering Inc., Norwalk, CT, USA). Subsequently, the beef samples were trimmed of subcutaneous fat and connective tissue, cut into 2 × 2 × 2.5 cm cubes, wrapped in aluminum foil, and randomly identified with three-digit codes. The samples were kept in a warm cabinet at 50 °C for less than 15 min and were submitted to sensory analysis.

### 2.4. Conjoint Analysis

Conjoint analysis is a multivariate research technique that assumes that purchasing behavior can be interpreted as a choice between different products or brands that have a set of differentiating attributes or characteristics. If the product alternatives can be defined by a set of specific levels of a common set of attributes, then the total utility of a product to a consumer is given by the partial utilities of each attribute level [63].

The aim of the conjoint analysis was to determine the relative importance of five attributes related to the purchase of beef in Spain: price, origin, organic certification, animal welfare certification, and protected geographical indication. These attributes were selected because they refer to very relevant aspects in beef production and consumption, and it was intended to verify their contributions to consumer purchase decisions [8,37,47,48,52,55,56,59,64,65,66,67,68,69,70,71,72,73,74,75,76,77,78,79,80,81,82,83,84,85].

Origin had two different levels: “national” and “EU imported”. These two levels were chosen to determine the importance of national beef in the purchase intention compared to the most common import, which is from EU countries.

Quality labels (OA, PGI, and WELFAIR™) had two levels, “presence” or “absence” of the label, in order to determine the importance of each of these certifications in comparison with the absence thereof, reflecting the decision-making circumstances usually faced by consumers.

Finally, price had three levels: 14, 18, and 22 EUR/kg. Prices reflected those found in the market a few weeks prior to the start of the study, and values were rounded to zero decimal places. The low price was set as the average price of nondifferentiated beef tenderloin, while the high price was set as the average price of beef tenderloin with at least one quality certification. The mean of these two prices was considered the medium price.

The combination of these 5 attributes and their 11 levels results in 48 different profiles. An orthogonal design was used to reduce the combinations to just eight in order to avoid fatigue and routine responses [84]. Therefore, each participant was given eight labels identified with a random code (Figure 1). Participants received detailed written descriptions of the different products to be evaluated and were asked to order the labels according to their purchasing preferences, from 1 (most preferred) to 8 (least preferred). Classification was chosen over scoring, as it has been reported to provide better results [86] and has been previously used as a criterion in other studies [8,59,61].

The inclusion of price in the conjoint analysis allows the estimation of the monetary value that consumers place on the presence of other attributes. The willingness to pay for a unit increase in an attribute was calculated by dividing the utility of each attribute other than price by the price coefficient [87]. In this way, the premium that consumers are willing to pay for each label (OA, PGI, and AW) was determined.

### 2.5. Data Analysis

Data analysis was performed using the statistical software SAS version 9.4 (SAS Institute Inc., Cary, NC, USA). Consumer preferences for the effects of diet and time of aging were analyzed using a generalized linear model (GLM) procedure for each attribute assessed. The fixed effects in the model were diet, time of aging, and the interaction between the two factors. Differences between least-squares means were obtained at *p* < 0.05 using the Student–Newman–Keuls test (SNK). Analysis was performed for occasional consumers, regular consumers, and the whole sample set. The overall assessment was also analyzed for different consumer segments according to age, gender, educational level, and income level.

Nonmetric conjoint data were analyzed using the TRANSREG procedure of SAS. The applied model considers the monotonic transformation as the sum of all partial utilities for each attribute equal to zero. This is a general and flexible model that is generally used for qualitative data. Although price is numerical, the aim was to include a low, a medium, and a high price, so it was considered as qualitative data for the analysis [8,86]. The relative importance of each factor was obtained, as well as the utility values associated with each level. The analysis was carried out for the entire sample and for each consumer segment according to age, gender, educational level, income level, and frequency of consumption of beef. The analysis was performed for occasional consumers, regular consumers, and the entire sample.

The relative importance and willingness to pay for each attribute were compared according to the consumer segment (consumption frequency, age, gender, educational level, and income level) using a nonparametric test (Kruskal–Wallis) with the NPAR1WAY procedure.

## 3. Results

### 3.1. Consumer Liking

Table 2 shows the results obtained from the sensory evaluation for all of the established attributes (statistical significance was declared at *p* < 0.05). The aging of the meat was found to be a more important factor than diet for the formation of sensory preferences in the three groups of consumers analyzed. All of the considered groups of consumers ranked the meat aged for 21 days as the best for all evaluated attributes, except for tenderness, whose best rating was obtained by meat aged for 28 days. However, no differences were observed within the group of occasional consumers. The different consumer groups did not differentiate the meat aged for 7 days from the meat aged for 28 days except when evaluating tenderness.

Occasional consumers did not report a sensory difference between finishing diets, while regular consumers gave a better sensory evaluation to the meat from animals on the alternative diet, considering it to be more tender and have better color and flavor. Similar results were obtained for the whole set of consumers, although for this group, differences in tenderness and general assessment only reached a trend level. No participants in any of the groups reported differences in juiciness or odor between diets.

Figure 2 represents the global evaluation of meat made by different consumer segments according to gender, age, educational level, and income level. The preference for 21-day aging was the tendency in most of the considered consumer segments, except for those over 65 years of age, who preferred meat aged for 21 days. In contrast, most consumer segments did not differentiate between different finishing diets, except for those over 65 years of age and university students, who preferred meat from animals on the alternative diet.

### 3.2. Conjoint Analysis

The relative importance and utility value of the five studied factors (price, origin, PGI, OA, and AW certification) for the groups of consumers are shown in Table 3. No significant differences were found between regular and occasional consumers.

The most important attribute for the formation of preferences was price (28.17%), followed by origin (25.01%) and AW certification (18.57%), with a marked preference for the lowest price, national origin, and the presence of the AW label. The attributes that least affected the choice of the different types of meat were OA certification and PGI, with a relative importance of 14.15% and 14.09%, respectively.

Figure 3 presents the relative importance of the attributes in the formation of preferences for different consumer segments according to gender, age, education level, and income level. The OA certification was significantly less important for the formation of preferences in participants with university degrees than in those with other educational levels. Price was significantly more important for those over 60 or under 25 years of age, while AW certification was significantly more important for consumers aged 25–40 and less important for those over 60 years old.

### 3.3. Willingness to Pay

Table 3 also displays the willingness to pay for the preferred level of the assessed attributes for regular consumers, occasional consumers, and the whole sample set. No significant differences were observed between regular and occasional consumers.

The highest willingness to pay corresponded to national origin compared to EU origin (8.18 EUR/kg), followed by AW certification (5.53 EUR/kg) and PGI (3.61 EUR/kg). The lowest willingness to pay corresponded to the OA certification (1.49 EUR/kg).

## 4. Discussion

The combination of sensory and conjoint analyses made it possible to evaluate preferences at the place of purchase (defined by extrinsic attributes—e.g., origin and price of the product) and those of actual consumption (based on intrinsic attributes—e.g., color, flavor, texture, or juiciness), which are the main factors responsible for the purchase decision and its future repetition [41].

### 4.1. Sensory Analysis

Results from the sensory analysis evidenced that the alternative diet based on raw materials and local by-products from southern Spain (CA) improved the general assessment of beef by regular consumers, while occasional consumers did not differentiate between diets. A lower frequency of consumption of beef can prevent the recognition of small variations in attributes based on experience [88], which could be the main cause of the lower discrimination capacity of occasional consumers.

These results complement those obtained by Moreno-Diaz et al. [17], who reported that the alternative diet did not present negative effects on growth performance or carcass and meat traits. However, it is possible that the alternative diet might have induced small variations in meat traits that affected sensory characteristics (such as color or flavor) and were not detected by Moreno-Díaz et al. [17] (i.e., due to sensitivity of the instrumental techniques, variability between experimental units, etc.) but did not pass unnoticed by regular users who have enough experience to detect small variations in sensory characteristics. In any case, information is now available so that the livestock sector can use it to finish calves and, in this way, strengthen the sustainability of the supply chain by creating value from raw materials and local by-products [89].

The color of cooked meat, which depends on the extent of ferrihemochrome formation, can be influenced by factors such as pH, meat source, packaging conditions, freezing history, fat content, and added ingredients [90], which change the ratio of different forms of myoglobin. In the present study, regular consumers assigned better ratings to the color of meat from the experimental batch, which could be attributed to the difference in the pH_24_ of meat between diets [17].

The juiciness scores were similar between diets, which agrees with the results obtained by Moreno-Diaz et al. [17] when analyzing the water-holding capacity (drip and cooking losses) in the same animals of the current study, although the connection between cooking loss and juiciness is not simple.

Odor and flavor are complex sensations. They are mainly due to the release of volatile substances from the degradation of meat components (lipids, carbohydrates, amino acids, etc.) [91,92], although nonvolatile compounds are also known to have an impact on flavor. While diet is included among the factors that can affect flavor [93], in the present study, consumers did not find differences in the odor of meat, but they did assign better ratings to the flavor of meat from animals fed the experimental diet. This is in line with the general perception that consumers give the worst ratings to meat from animals on compound feeds versus more fibrous feeds [94]. As there are no differences in the pH_24_ of the meat [17], and therefore in lactic acid content (another precursor of aromatic compounds), this difference can be attributed to the content of unsaturated fatty acids (especially linolenic acid), as these are the main precursors of the flavor of cooked meat [95]. Nevertheless, the obtained results suggest that consumers did not detect any undesirable flavor associated with high levels of linolenic acid. It is well known that the relative levels of R-linolenic acid (C18:3 n-3) are largely responsible for differences in volatile composition and, hence, the flavor of beef [96]. In sheep fed the same diets as in the present work [97], the meat of animals fed with the alternative diet showed higher contents of monounsaturated fatty acids (MUFAs) and linolenic acid. However, some authors [98] reported that an excess of polyunsaturated fatty acids (PUFA) could have a negative impact on the characteristics of flavor.

According to Brooks et al. [99], toughness is the main determinant of consumer satisfaction regarding beef. Bovine meat toughness is a complex property that depends mainly on the connective tissue and myofibrils. Contrary to what was expected based on the results obtained by Moreno-Diaz et al. [17], who did not find differences between diets for the toughness of meat cuts from the same animals used in the present study, regular consumers assigned better ratings to meat from animals on the experimental diet. The differences were negligible for occasional consumers. These results may have a positive influence on new purchase decisions. The mean values of the shear force were 5.73 and 5.43 kg/cm^3^, which ranks the meats between the tough and tender categories according to the scale established by Destefanis et al. [100], who indicated that beef with Warner–Bratzler (WB) shear force values of >52.68 and <42.87 N is perceived by most consumers as ‘tough’ and ‘tender’, respectively. The number of untrained panelists should not be considered the reason for these divergences as, according to Wheeler et al. [101], the correlation of mean panel ratings to shear force is acceptable when the number of panelists is greater than 16 [101]. Thus, we could attribute these deviations to differences in pH_24_ [102]. In a sensory evaluation of meat from two production systems (intensive vs. free-range organic), consumers evaluated the meat from a free-range organic system as harder [84], which was attributed to the extensibility of the system.

Wet aging is the most dominant packaging method in the current meat industry due to associated advantages regarding economic (significant reductions in product weight and trim loss), production (less space required, adaptable to automation, and efficient product flow), and microbial factors (extended shelf-life without sacrificing palatability traits) [30]. However, aging is a great expense for the meat industry, so the aging time should not be prolonged more than necessary to achieve the level of sensory quality appropriate to the profile of consumers [103].

The impact of aging on the quality of beef has been previously investigated in many experiments, most of which have shown a beneficial effect, mainly in tenderness [104]. In this regard, Gorraiz et al. [105] indicated that 7-day aging of beef yielded an increase in characteristic flavor and aftertaste intensity, causing an appreciable improvement of its flavor. In beef aged for 1 and 7 days, Ornaghi et al. [106] and Torrecilhas et al. [107] reported that aging improved odor, flavor, tenderness, and overall acceptability. However, unnecessarily long aging times can adversely affect the sensory quality of meat [108]. Based on the results from the present study, aging time had a greater influence than diet on the sensory attributes of meat. For all of the studied attributes except for tenderness, the best ratings were assigned to meat aged for 21 days, and no differences were found between meat aged for 7 and 28 days. The improvement of the sensory attributes of meat may be related to the formation of volatile compounds from enzymatic changes during aging [109].

Differences in juiciness could be related to the water-holding capacity of fresh meat and its reduction in meats aged for 21 days. In contrast to these results, some authors [110] reported that juiciness was not affected by the aging period (42 days). The differences observed in odor and general acceptance could be attributed to the degradation of proteins during aging and the appearance of undesirable flavors after 21 days of aging.

There were changes in the pH and content of different forms of myoglobin in the fresh meat from all studied animals [17]. These two factors are the main ones responsible for the differences found in the consumers’ assessment of the color of cooked meat in the present study. Myoglobin oxygenation can be altered by even small changes in the length of time that a product has been packaged [111]. Likewise, some authors [112] indicated that myoglobin has an important role in the oxidation of polyunsaturated lipids. While the oxidation processes involved in meat cooking will alone increase MetMb and ferrihemochrome, the effect could be increased if the fat content is higher. Some studies [113] verified the effect of the form of myoglobin on the color of cooked meat, which was also established by other authors [114] for ground and whole-muscle pork.

Tenderness improved with aging, while some authors found no improvement in tenderness beyond day 14 [115,116] or beyond day 21 [117,118]. There is no agreement in the results from the literature. Some authors [119] did not find significant differences in the acceptability, juiciness, or flavor of meats aged from 2 to 63 days, while tenderness only seemed to improve until 14 days of aging. For an M. gluteus medius cut of meat with different storage times (0, 15, and 30 days), a recent study [120] reported that all sensory scores (color, taste, texture, juiciness, odor, and acceptability) declined, with the lowest rates observed at day 30. Garmyn et al. [32] observed that for beef aged for 21–84 days, aging time did not influence tenderness, while juiciness, flavor, and overall eating quality diminished with an increase in aging. In the present study, participants did not differentiate between different aging periods, although the general trend was a decrease in acceptance with the advancement of aging, with consumers being more satisfied with 35-day aged meat than with meat aged for 63 or more days. These divergences could be related to changes in the levels of volatile compounds originating in intramuscular lipid degradation [105]. The decrease in juiciness with aging time is attributed to the loss of water during this process, while variations in odor and flavor could be due to the appearance of off-flavor attributes and the attenuation of desirable flavors [17], although storage temperature can also play an important role. In particular, desirable flavor intensity increases at a temperature of 1 °C, while off-flavor intensity rapidly increases at 5 °C [17].

### 4.2. Conjoint Analysis and Willingness to Pay

From the conjoint analysis and based on the estimated utility for each level of the evaluated attributes, the preferred meat, both for regular and occasional consumers, is that of national origin, at the lowest possible price, and including all possible quality labels. These results are consistent with most of the studies that assess extrinsic values in the beef purchasing intention [59,64,84,85]. The relative importance of the assessed attributes was similar in both occasional and regular consumers, which contrasts with previous studies that usually reported changes in relative importance according to the level of meat consumption [48,58]. The general trend reported is that regular consumers pay more attention to the region of origin, while quality labels are more important for occasional consumers [121,122]. The present study did not consider the region of origin, which makes comparison with results from previous studies difficult.

Price and origin have been the most traditionally evaluated extrinsic factors of food products [37,42]. Both attributes have usually been significant for beef purchasing decisions, although the relative importance of both factors has changed over time with opposing trends as a result of greater consumer interest in the quality of meat and its possible health effects [64].

Previous studies have shown that the importance of price for consumers has been decreasing from the almost 50% reported by Sanchez et al. [65] to the around 25% found in the most recent studies [42,84]. Some authors [66] reported that Spanish consumers are generally much more sensitive to the price of beef than to the price of lamb, with a relative importance of this factor between 5 and 33%, depending on the consumer segment. A study performed in 2016 [84] found that the relative importance of price was 26.2% for a group consisting mainly of older consumers with university degrees. This agrees with the results from our work, where consumers over 60 years of age and consumers with university degrees had a higher sensitivity to this factor. In any case, in the present study, price always had a negative utility, so the preferred level was the lowest, which is consistent with most references from other authors [42,64,84].

Regarding the origin of the meat, the literature reveals that for Western consumers, origin is one of the most important attributes when purchasing beef [37,59,123]. According to Mesias et al. [64], the relative importance of the national origin compared to imported meat has increased, and national origin is the usual choice for both Spanish and European consumers [42,84]. This preference for national production has been related to the pursuit of food safety based on a known origin [67,84], social acceptance, and support for national producers [68]. The fact that meat is of national origin seems to be considered an element that guarantees quality [48] and strengthens the sense of belonging [69]. Previous research has also proved that consumers are willing to pay more for beef of national origin [123]. In the present study, the highest premium that consumers were willing to pay was for national origin, followed by the AW label.

Quality labels provide indications of different aspects that impact purchasing behavior [48,85]. These labels are of great relevance for the consumer to form their quality expectations, as generally, at the time of purchase, it is difficult to recognize the intrinsic attributes of meat that provide quality to the product [70]. Moreover, the perception of quality and purchasing preferences can also vary with time [71].

European consumers are increasingly paying more attention to health, food safety, and ethical factors related to production systems, among others, leading to a growing proliferation of quality labels that provide indications of these aspects [37,59]. Quality labels are especially relevant for the beef sector, where consumers have lost confidence in meats of unknown origin and seek references that indicate control and security [64]. However, some studies have highlighted that the excessive use of quality labels leads to confusion and can undermine the value that consumers place on them [72,85]. According to a previous study, consumer-led product development should incorporate emerging quality attributes that are relevant to an increasing number of consumers [101]. Hence, the importance of developing differentiation strategies is led by the impact of different types of information on consumer behavior [124].

The three quality labels evaluated in this study had increased utility for consumers, although none of them achieved a higher utility than that obtained by national origin. This is in line with some previous studies that have evidenced that the origin of meat represents a better guarantee of quality than that offered by the PGI and OA labels [47]. According to Bernabeu et al. [48], quality labels seem to constitute secondary criteria of preference after national origin, which could accelerate the purchase of meat compared to the more detailed analysis required when considering quality labels.

Previous studies have also proved that the importance of the PGI level may vary with the geographic location of consumers, and its impact on purchase decisions can differ between regional groups due to the fact that consumers from producing regions would be more interested in meat produced locally [125]. The present study was carried out in three areas with no productive links with any PGI. However, it is possible that the relative importance of the PGI label and the premium that the consumer is willing to pay for it may increase in producing regions linked to a particular PGI label.

AW is the most valued quality label and the one for which consumers are willing to pay a higher premium. Animal welfare is an attribute that has gained a high importance for consumers in the last few years [52,56]. A decade ago, animal welfare was perceived as a desirable condition, but consumers were not willing to pay much more for better animal welfare [126,127]. According to several authors [55,73], consumers prefer to buy animal-welfare-friendly products, as they consider there to be an association between animal welfare at a farm level and higher benefits for human health. Other studies have reported that consumers also consider animal-welfare-friendly products as higher quality, tastier, more hygienic, and safer [128,129]. However, animal welfare labels are of recent creation and are not currently endorsed by an official EU certification scheme, so consumers might tend to simplify the complexity and multidimensionality of the concept of animal welfare [74].

Different studies have shown that the interest of consumers in organic meat arises mainly from altruistic motivations, highlighting the beneficial effect of organic production on the protection and conservation of the environment [75,76]. Moreover, consumers are also interested in the greater health benefits derived from higher animal welfare standards and the low use of pesticides and antibiotics [77,130]. However, the OA label is less valued than the AW label, which may be due to the limited knowledge that consumers have about some of the expected benefits previously described [78,84]. According to [79], this could also be related to a certain lack of confidence of some consumers in the controls that guarantee organic certifications. If consumer interest in organic beef is to be strengthened, communication of the aspects that can motivate their purchase (e.g., food safety and environmental impact) should be improved [76,80], and the certification process for OA needs to be better disclosed to build more confidence in the label [81].

In any case, the premium that the consumer is willing to pay for the OA label is considerably lower than the high price of organic production, which is consistent with previous studies [50,76,82]. Consequently, price continues to be an important barrier to increasing the consumption of organic beef.

On the other hand, there is some overlap between the three quality labels considered in the present study. This is due to the fact that they refer to certain conditions and characteristics of the production system that are not usually available to the consumer at the time of purchase. Furthermore, consumers do not possess enough knowledge to have a complete understanding of each of these labels [8,74]. The overlap between the OA and AW labels is especially relevant, as protection of animal welfare is also considered a basic aspect of organic agriculture, so in this sense, both labels are in competition. The results evidence that consumers attribute a higher value to beef with the AW label than to organic beef and are also willing to pay a higher price for the AW than for the OA label. As the retail price for organic beef is higher, consumers are more likely to buy beef with an AW label. Similar results were recently reported by Akaichi et al. [83] for bacon. These findings suggest that consumers do not have a full understanding of the higher level of animal welfare of organic agricultural systems and, consequently, will be less willing to pay a premium for organic beef.

There are, however, two main limitations to this work. First, consumer studies were carried out in three cities from the southeast of Spain, while the results were extrapolated to the country as a whole. Secondly, the contrary trend to the results reported in the literature for consumer choices regarding organic beef could indicate a bias in the consumer sample. Both of these issues might have influenced consumer responses.

## 5. Conclusions

Sensory analysis of beef revealed that, in experimental conditions, aging is a more relevant factor than the finishing diet. Meat aged for 21 days obtained the best scores for all of the evaluated sensory attributes and overall assessment, except for tenderness, where the best rating was obtained by meat aged for 28 days. Feeding a finishing diet rich in agro-industrial by-products, mostly fibrous, improves the color, taste, tenderness, and overall assessment by regular consumers and does not affect the evaluation by occasional consumers.

Conjoint analysis revealed that national origin and the lowest price continue to be the attributes that most determine the willingness of Spanish consumers (both regular and occasional) to purchase beef, while quality labels have less influence on preferences. There is an increase in utility and economic value that consumers attribute to meat obtained in systems that ensure animal welfare, which exceeds that of meat obtained in organic agriculture systems or with a protected geographical indication. Producers are encouraged to focus on ensuring animal welfare to improve sales opportunities.

## Figures and Tables

**Figure 1 animals-12-00006-f001:**
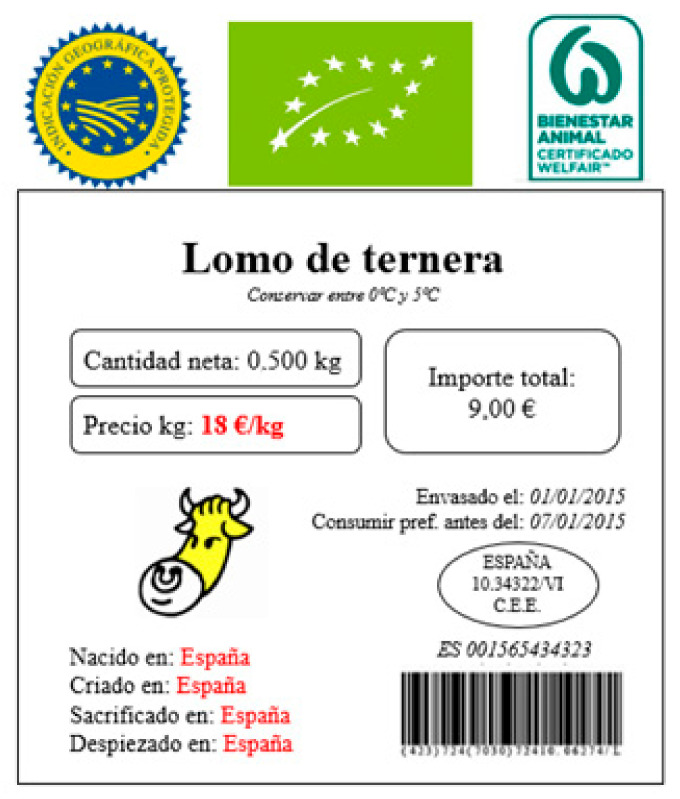
An example of a beef label presented in the conjoint analysis.

**Figure 2 animals-12-00006-f002:**
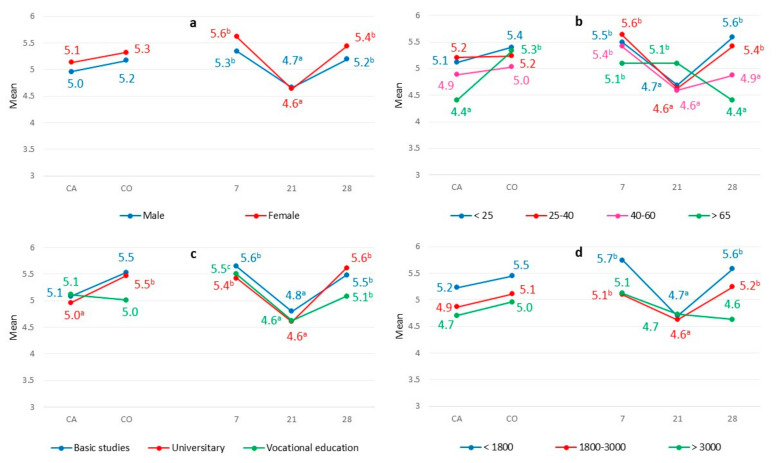
Overall assessment from different feeds (CA, by-product-based diet; CO, conventional feed) and aging (7, 14, and 21 d) for consumer segments according to gender (**a**), age (**b**), educational level (**c**), and income level (**d**). Means with different superscripts are statistically different (*p* < 0.05).

**Figure 3 animals-12-00006-f003:**
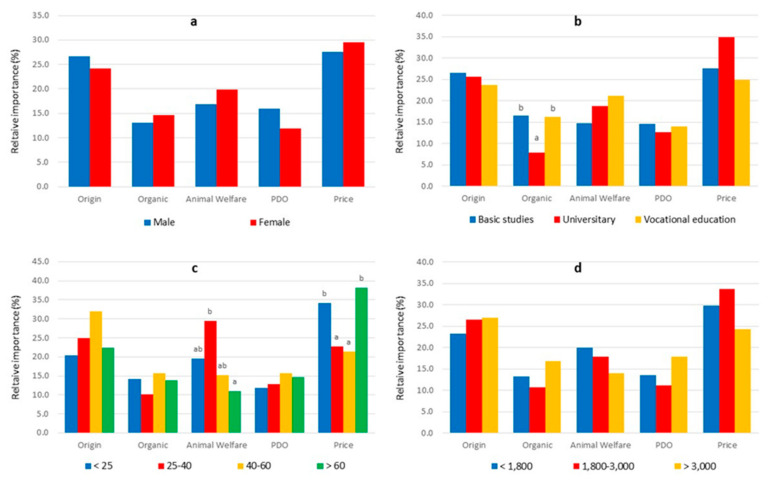
Relative importance (%) of the attributes for consumer segments according to gender (**a**), educational level (**b**), age (**c**), and income level (**d**). Means with different superscripts are statistically different (*p* < 0.05).

**Table 1 animals-12-00006-t001:** Demographic characteristics of consumers and the Spanish population.

	Sample	Spanish Distribution ^a^
Age group		
18–25	27.35	26.07
25–40	18.28	18.76
40–60	33.01	28.94
60–76	21.36	26.23
Gender		
Male	49.51	49.02
Female	50.49	50.98
Educational level		
Basic education	34.62	33.64
University education	35.58	38.60
Vocational education	29.81	22.75

^a^ Source: INE 2019 [57].

**Table 2 animals-12-00006-t002:** Sensory assessment of beef with different feeds and aging for all consumers, regular consumers, and occasional consumers of beef.

Consumer	Variable	Feed			Aging				Feed × Aging							SEM
		CA	CO	F	7 d	21 d	28 d	F	CA 7 d	CA 21 d	CA 28 d	CO 7 d	CO 21 d	CO 28 d	F	
Global	Color	5.04 ^a^	5.28 ^b^	3.59	5.52 ^b^	4.58 ^a^	5.38 ^b^	21.46	5.10 ^b^	4.56 ^a^	5.47 ^b^	5.95 ^c^	4.61 ^a^	5.30 ^b^	6.10	0.07
	Juiciness	4.87	5.00	0.84	5.28 ^b^	4.49 ^a^	5.04 ^b^	9.81	4.67 ^b^	4.82 ^b^	5.11 ^b^	5.89 ^c^	4.16 ^a^	4.97 ^b^	14.06	0.08
	Tenderness	4.93	5.19	3.25	5.46 ^b^	4.54 ^b^	4.17 ^a^	14.29	4.83 ^b^	4.66 ^b^	4.31 ^ab^	6.10 ^c^	4.42 ^b^	4.03 ^a^	12.37	0.08
	Odor	5.24	5.31	0.37	5.55 ^b^	4.83 ^a^	5.46 ^b^	13.12	5.41	4.74	5.58	5.69	4.92	5.34	1.54	0.06
	Taste	5.18 ^a^	5.50 ^b^	5.32	5.67 ^b^	4.81 ^a^	5.54 ^b^	15.27	5.21 ^ab^	4.74 ^a^	5.60 ^c^	5.21 ^ab^	4.89 ^a^	5.48 ^b^	5.11	0.07
	Overall	5.06	5.26	3.01	5.50 ^b^	4.65 ^a^	5.34 ^b^	20.54	5.05 ^ab^	4.70 ^a^	5.44 ^b^	5.96 ^c^	4.60 ^a^	5.23 ^b^	9.59	0.06
Regular consumers																
	Color	4.93 ^a^	5.24 ^b^	4.88	5.52 ^b^	4.50 ^a^	5.23 ^b^	18.55	4.98 ^bc^	4.52 ^ab^	5.29 ^c^	6.06 ^d^	4.48 ^a^	5.17 ^c^	7.57	0.08
	Juiciness	4.72	4.95	1.70	5.17 ^b^	4.43 ^a^	4.90 ^b^	6.29	4.50 ^ab^	4.85 ^b^	4.83 ^b^	5.85 ^c^	4.02 ^a^	4.98 ^b^	13.34	0.09
	Tenderness	4.87 ^a^	5.20 ^b^	4.37	5.51 ^c^	4.50 ^b^	4.10 ^a^	13.39	5.17 ^bc^	4.65 ^ab^	4.09 ^a^	6.15 ^c^	4.35 ^ab^	4.11 ^a^	9.13	0.09
	Odor	5.26	5.35	0.41	5.61 ^b^	4.85 ^a^	5.47 ^b^	10.07	5.46	4.74	5.59	5.76	4.96	5.35	1.31	0.08
	Taste	5.08 ^a^	5.47 ^b^	6.49	5.63 ^b^	4.78 ^a^	5.43 ^b^	11.04	5.11 ^ab^	4.76 ^a^	5.38 ^b^	6.16 ^c^	4.80 ^a^	5.47 ^b^	4.50	0.08
	Overall	4.98 ^a^	5.25 ^b^	4.03	5.49 ^b^	4.61 ^a^	5.24 ^b^	15.86	4.98 ^bc^	4.70 ^ab^	5.27 ^c^	6.00 ^d^	4.52 ^a^	5.22 ^c^	8.30	0.07
Occasional consumers																
	Color	5.40	5.42	0.01	5.53 ^b^	4.83 ^a^	5.87 ^b^	4.97	5.47 ^abc^	4.67 ^a^	6.07 ^c^	5.60 ^abc^	5.00 ^ab^	5.67 ^bc^	0.64	0.14
	Juiciness	5.31	5.18	0.22	5.60 ^b^	4.67 ^a^	5.47 ^b^	4.23	5.20 ^ab^	4.73 ^ab^	5.94 ^b^	6.00 ^b^	4.60 ^a^	4.93 ^a^	3.61	0.15
	Tenderness	5.13	5.15	0.01	5.33	4.70	4.40	2.97	4.73	4.67	4.48	5.93	4.73	4.38	4.98	0.16
	Odor	5.18	5.20	0.01	5.34 ^b^	4.77 ^a^	5.43 ^b^	3.06	5.27	4.73	5.53	5.47	4.80	5.33	0.24	0.12
	Taste	4.49	5.58	0.09	5.80 ^b^	4.90 ^a^	5.90 ^b^	4.59	5.53 ^abc^	4.67 ^a^	6.27 ^c^	6.07 ^bc^	5.13 ^ab^	5.53 ^abc^	1.92	0.15
	Overall	5.30	5.30	0.00	5.53 ^b^	4.78 ^a^	5.61 ^b^	5.22	5.24 ^ab^	4.69 ^a^	5.97 ^b^	5.81 ^b^	4.85 ^a^	5.25 ^ab^	2.67	0.12

Means with different superscripts (a, b, c, d) are statistically different (*p* < 0.05); CA (by-product-based feed), CO (conventional feed); F (F-ratio); d (days of ageing).

**Table 3 animals-12-00006-t003:** Relative importance (%), utility values, and changes in willingness to pay (EUR/kg) for each attribute for all consumers, regular consumers, and occasional consumers of beef.

Variable	Global	Consumer	
		Regular	Occasional
Intercept	4.55	4.50	4.69
Origin			
Spain	0.77	0.75	0.82
EU imported	−0.77	−0.75	−0.82
Relative importance (%)	25.01	25.75	22.92
Willingness to pay (EUR/kg, Spain)	8.18	8.01	8.59
Organic label			
Yes	0.14	0.13	0.17
No	−0.14	-0.13	−0.17
Relative importance (%)	14.15	13.53	15.92
Willingness to pay (EUR/kg, Yes)	1.49	1.39	1.78
Animal welfare label			
Yes	0.52	0.46	0.68
No	−0.52	−0.46	−0.68
Relative importance (%)	18.57	17.92	20.42
Willingness to pay (EUR/kg, Yes)	5.53	4.91	7.12
PGI label			
Yes	0.34	0.31	0.44
No	−0.34	−0.31	−0.44
Relative importance (%)	14.09	14.32	13.45
Willingness to pay (EUR/kg, Yes)	3.61	3.31	4.61
Price (EUR/kg)			
14	0.72	0.75	0.65
18	0.06	0.00	0.23
22	−0.78	−0.75	−0.88
Relative importance (%)	28.17	28.48	27.27
R^2^	0.93	0.93	0.95

## Data Availability

The data presented in this study are available on request from the corresponding author. The data are not publicly available due to project IP rules.

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
