# Peer review of "Beef from Calves Finished with a Diet Based on Concentrate Rich in Agro-Industrial By-Products: Acceptability and Quality Label Preferences in Spanish Meat Consumers"

_animals, 2021, doi:10.3390/ani12010006_

Round 1

Reviewer 1 Report

Dear Authors,

the manuscript has been significantly improved. However, you should definitely reflect in the limitation of the study on the contradictory as compared with the existing literature results on the relation between education level and relative importance of attribute “organic” (Figure 3). Make it explicit what the limitations of the your study are and what they arise from.

Author Response

Dear Reviewer 1

We appreciate the feedback and all the recommendations received. As suggested, we have included a paragraph at the end of the discussion, where we report those limitations pointed out by Reviewers 1 and 2 regarding the location of the study and the relative importance of the attribute “organic” in relation to the literature. Please see this below: 
“There are, however, two main limitations to this work. First, consumer studies were carried out in three cities from the southeast of Spain, while results have been extrapolated to the country as a whole. Secondly, the contrary trend to what was reported in the literature for consumer choices regarding organic beef could indicate a bias in the consumer sample. Both these issues might have influenced consumer responses.”
We now believe that the manuscript is ready to be accepted for publication. 

Kind regards, 
Dr. Francisco Requena Domenech

Reviewer 2 Report

I still have big concerns about the interest of the work. I consider that this is limited, due to its location in a small area of Spain and that it provides little data, as it only uses previous references to other work.
However, the authors have addressed some of the previous comments.

Author Response

Dear Reviewer 2

We appreciate the feedback and all the recommendations received. As suggested, we have included a paragraph at the end of the discussion, where we report those limitations pointed out by Reviewers 1 and 2 regarding the location of the study and the relative importance of the attribute “organic” in relation to the literature. Please see this below: 
“There are, however, two main limitations to this work. First, consumer studies were carried out in three cities from the southeast of Spain, while results have been extrapolated to the country as a whole. Secondly, the contrary trend to what was reported in the literature for consumer choices regarding organic beef could indicate a bias in the consumer sample. Both these issues might have influenced consumer responses.”
We now believe that the manuscript is ready to be accepted for publication. 

Kind regards, 
Dr. Francisco Requena Domenech

Reviewer 3 Report

I have reviewed this manuscript before. There were only minor revisions necessary which have been fully included in the new version. I do not see further reasons to deny publication of the manuscript.

Author Response

Dear Reviewer 3, 

Thanks so much for your wonderful comments. 

Dr. Francisco Requena